# Pesticide Residues Identification by Optical Spectrum in the Time-Sequence of Enzyme Inhibitors Performed on Microfluidic Paper-Based Analytical Devices (µPADs)

**DOI:** 10.3390/molecules24132428

**Published:** 2019-07-02

**Authors:** Ning Yang, Naila Shaheen, Liangliang Xie, Junjie Yu, Hussain Ahmad, Hanping Mao

**Affiliations:** 1School of Electrical and Information Engineering, Jiangsu University, Zhenjiang 212013, China; 2School of Agricultural Equipment Engineering, Jiangsu University, Zhenjiang 212013, China

**Keywords:** organophosphorus and carbamates pesticide, pesticide residues rapid detection, time detection model

## Abstract

Pesticides vary in the level of poisonousness, while a conventional rapid test card only provides a general “absence or not” solution, which cannot identify the various genera of pesticides. In order to solve this problem, we proposed a seven-layer paper-based microfluidic chip, integrating the enzyme acetylcholinesterase (AChE) and chromogenic reaction. It enables on-chip pesticide identification via a reflected light intensity spectrum in time-sequence according to the different reaction efficiencies of pesticide molecules and assures the optimum temperature for enzyme activity. After pretreatment of figures of reflected light intensity during the 15 min period, the figures mainly focused on the reflected light variations aroused by the enzyme inhibition assay, and thus, the linear discriminant analysis showed satisfying discrimination of imidacloprid (Y = −1.6525X − 139.7500), phorate (Y = −3.9689X − 483.0526), and avermectin (Y = −2.3617X − 28.3082). The correlation coefficients for these linearity curves were 0.9635, 0.8093, and 0.9094, respectively, with a 95% limit of agreement. Then, the avermectin class chemicals and real-world samples (i.e., lettuce and rice) were tested, which all showed feasible graphic results to distinguish all the chemicals. Therefore, it is feasible to distinguish the three tested kinds of pesticides by the changes in the reflected light spectrum in each min (15 min) via the proposed chip with a high level of automation and integration.

## 1. Introduction

Pesticides play a crucial role in increasing crop yields worldwide due to their significance in decreasing the number of insects and diseases [1]. In some developing countries, pesticides are estimated to be used excessively [2]. In the developed world, this issue is not trivial either; for example, over 1,000,000 pounds of pesticides are estimated to be used in America each year [3]. Thus, pesticide exposure is quite problematic globally [3], and pesticide residues in food may lead to the deterioration of food safety and public health [4]. Even minimal exposure to pesticides in daily intake of food can arouse health effects like headaches and many types of cancer in humans [5], which include lymphocytic and hematopoietic liver, breast, and ovarian cancer [6]. Other health risks like non-Hodgkin’s lymphoma or leukemia can also occur because of pesticide residues in food [5]. Different kinds of pesticides vary in the level of poisoning, among them, methyl parathion (*O*,*O*-dimethyl *O*-(4-nitrophenyl) phosphorothioate), avermectin, and phorate are deemed as highly toxic, with adverse effects including abnormal development of the embryo during pregnancy or hyperglycemia [7]. By contrast, pesticides like imidacloprid are much more harmless.

Therefore, various techniques are proposed to tackle this health issue. Ye et al. used high-performance liquid chromatography (HPLC) to analyze chiral pesticides [8]. Jon et al. used a capillary gas chromatography and mass chromatography (GC/MS) method to achieve the multi-analysis of several pesticides [9]. These methods are powerful, efficient, and sensitive while retaining the ability to discriminate pesticides [10]. However, these methods require considerable amounts of regents, trained technicians, and time-consuming preparation procedures, resulting in an inconvenience in low-resource settings. With regard to other methods, Vetrova et al. used a bioluminescent signal system to indicate toxicants like organophosphates [11]. Zhao et al. (2012) successfully used interactions, including van der Waals forces and hydrophobic forces, to build quantum dots-based molecularly imprinted polymers (QDs-MIPs), which can be used for specific recognition and fluorescent analysis for pesticides [12]. In addition, performance of biosensor pesticide residue detection technology is also satisfying [13]. Surface-enhanced Raman spectroscopy (SERS) makes the detection of non-fluorescent targets readily accessible, and subsequently, it is also effective in probing pesticide residues [14], although it is demanding to maneuver the aggregation level or the size of experiment molecules, leading to an undesirable precision level [14]. As for these methods, the strength of being rapid and sensitive in probing specific modules is attractive [11,12,13,14], but the disadvantages are similar to the above techniques. The performance of microfluidic analytical devices on medical and fabrication scenarios is eye-catching due to their strengths of being portable, reagent-efficient, inexpensive, integrated, rapid in analysis, and power-effective [15], which make them an ideal solution for public health control even in low-resource settings. Therefore, abundant pesticide assays are performed on microfluidic devices [16,17,18,19,20]. Duford et al. proposed a centrifugal microfluidic device which enables the integration of multiple processing procedures of enzyme inhibition-based determination while retaining the simultaneous detection of multiple samples or replicates in a single assay [16]. Furthermore, in 2018, Zhang et al. proposed a fluorescent pesticide determination system on a microfluidic paper chip, which makes ordinary material the protector against health hazards, providing a low-resource solution for the ubiquitous need for food protection [17].

Among microfluidic analytical platforms, enzyme inhibitors are the prevailing solution for pesticides determination [16,17,21]. Organophosphate pesticides (Ops) and carbamate pesticides (Cps) detection is based on the activity of the acetylcholinesterase enzyme (AChE) [18]. The presence of Ops and Cps inhibits the catalysis of AchE, and therefore, indophenol acetate barely reacts with H_2_O to produce acetic acid as well as indophenol (blue). It is apparent that the colorimetric readout provides a straightforward message which can be understood by almost every individual, while retaining a straightforward operation process [22,23]. However, this method only provides a basic solution to indicate the presence of pesticides, which lacks the ability to distinguish each pesticide. Optical spectroscopy can indicate the radiation loss of energy to the tested sample on a molecular scale [24,25]. Furthermore, it is non-invasive and imposes no ionizing radiation, thus, optical methods can capture the molecular activity in different chemical reactions and senses the energy involved [26]. Different inhibitors (pesticides) involved in the enzyme inhibition assays can lead to differences in chromogenic intensity and the number of acetylcholinesterase molecules inhibited at a specific time, which can lead to various optical figures if optical spectroscopy is used to track the whole chemical reaction. As mentioned above, different pesticides vary in the level of poisonousness. When those highly toxic chemicals just reach the detection limit or above, people tend to be misled by the equivocal chromogenic pattern. Thus, a misleading image of food safety is produced, as at this dose level, the health impair is likely to occur in vivo [27,28]. It is estimated that over 900,000 people lose their life due to the ingestion of highly toxic pesticides [29]. For this reason, governments assign various rules to regulate the crop market retailing and food industry. In China, the maximum residue level (MRL) for imidacloprid is 0.5 mg/Kg, while the MRL value for highly toxic pesticides like phorate or avermectin is both 0.01 mg/Kg [30], imposing greater sensitivity requirements in ubiquitous food security control. Therefore, it is natural to sense the chromogenic molecules’ activity and sequenced optical spectrum in the entire reaction process to distinguish the pesticide genre.

To solve these problems, a straightforward one-step microfluidic paper chip for pesticide identification was specially designed based on enzyme inhibitors and chromogenic molecules. This chip integrated the identification system by reflected light intensity in a time sequence, and then the fluctuation of radiation energy in a specific time could be captured via the proposed chip. Therefore, it can efficiently recognize different pesticides with greatly reduced manual operation. The proposed platform paves a potential way for ubiquitous pesticide identification, which gives comprehensive information and prewarning about daily-intake food.

## 2. Results and Discussion

### 2.1. Wavelength Selection (for Manual Analysis)

Different types of wavelengths can be used for the reference object; however, as shown in Appendix A, the wavelengths near 611.59 nm showed the greatest reflected light intensity, and thus, the recorded change was especially significant at this point, while showing the greatest linearity. Therefore, it was natural to choose this wavelength to estimate whether the reflected light intensity in a 15-min time sequence demonstrated different optical patterns by the various types of pesticide solutions tested. Subsequently, the captured principle can be further and automatedly analyzed by computers, and thus, attached to every pesticide with its unique optical identification card by the reflected light spectrum during the whole reaction period.

### 2.2. Averaged Time-Change Rate of Pesticides

Figure 1A demonstrates the optical experimental figures for three different pesticides, namely, avermectin, phorate, and imidacloprid. The peak light intensity of every min (at 611.59 nm) of nine tested chips was averaged, and is shown in the figure. It was obvious that the averaged reflected light intensity of avermectin was considerably lower than that of phorate and imidacloprid, which ranged from 468 to 1024 cd/m^2^. In addition, the distinct property of avermectin lies in the pattern of the line chart. The reflected light intensity decreased marginally without apparent fluctuation.

The line patterns of phorate and imidacloprid demonstrated resemblance, to some extent, with figures ranging from around 600 to 1200 cd/m^2^. However, the light intensity figures for phorate dropped significantly at the 3rd minute, and subsequently, made a small climb during the following minute, while the figures for imidacloprid decreased continuously during those two minutes.

In conclusion, the time sequence pattern of the various pesticides showed some clear distinctive properties, and it was estimated through cluster analysis and logarithmical modeling of reflected light intensity from chromogenic molecules and enzyme inhibitors that the pesticides can be classified by genera

### 2.3. Repetitive Assays

To verify the estimated property of the time sequence model of the reflected light spectrum of each pesticide, five of the proposed chips were prepared on a GDANA HT-300 digital heating device and numbered (No. 1 to 5). After 2 min, the No. 1 chip was introduced to the sample with an interval of 30 s. Then, the time counting began. At (*i* × 60 + (*No*. – 1) × 30) s, like the nine-chip assay, the corresponding chip’s optical information was recorded, until the time arrived at the 15th optical spectrum of the No. 5 chip.

As illustrated by Figure 1B, which was tested by 0.01 mg/Kg of avermectin, apart from one sample represented by black line, whose figures are relatively higher than the other four samples from 2 to 11 min. Overall, the values recorded by the other four chips fluctuated from about 480 to 1100 cd/m^2^; the figures and the former nine-chip assay were alike. Meanwhile, the black line did not show an ideal smoothness; this abnormal value may have been the influence of the manual operation of that chip. The enzyme inhibition assay contributed to the main chromogenic compound product in this chip, which affected the reflected light compared with the fabricated white color zone in the 7th layer. With the time flow, the inhibition activity tended to be steady and constant, which can lead to the smoothness of the graphic.

As for Figure 1C, the light intensity of the reflection of four chips decreased at the 3rd min, and then increased again. Although the remaining one did not show a similar pattern, the light intensity of it decreased at the 4th min, one min later than the other, and then the figures climbed again. Therefore, this feature is in harmony with the observed pattern of Figure 1A. The sudden decrease and increase in reflected light represented the unstable phorate-inhibited chromogenic compound at this time point, which is a distinctive property as well.

In the repetitive assay of imidacloprid, as shown in Figure 1D, all of the five tested chips showed a high level of smoothness, and the light intensity all ranged from 600 to 1300 cd/m^2^, which differed from those of avermectin. Thus, via repetitive assays, the distinctiveness in the optical spectrum model of different pesticides can be verified.

### 2.4. Identification of Pesticide by Time-Sequence Optical Spectrum Model

Different pesticide molecules lead to the different inhibition efficiencies of the same enzyme (AChE). Based on the former experiments, different enzyme inhibitors lead to the various optical spectrum pattern. Therefore, to establish the specific identification system for every tested pesticide, the original data came from the nine tested chips, as shown in Figure 1A; however, the entire optical spectrum was analyzed and not just the peak light intensity of a 611.59 nm wavelength. The completed optical spectrum for each min was 2047-dimension figures, which was undesirable for analysis. In addition, due to the operating environment and manual maneuvering, noise might occur in the statistics, which made the pretreatment of figures indispensable. Multiplicative scatter correction was used to purge the noise points. Then, for further accuracy, translation processing was used to improve the performance of the pesticide identification system. Principal information was distinguished by principal component analysis, and thus, the original 2047 dimension statistic was transformed into 20 dimension figures while maintaining the main optical features of every min, which made the analysis more straightforward. The discrimination results after linear discriminant analysis are demonstrated in Figure 2.

It is apparent that phorate can easily be distinguished from avermectin and imidacloprid, because its light intensity dropped significantly at the 3rd min and then made a noticeable climb again, which was different from the patterns of the other pesticides, which showed an overall continuous decreasing trend throughout the period. As for imidacloprid and avermectin, the tested figures showed a high extent of convergence with different pesticides, which means the digital distance among the pesticide genera can easily distinguish their identification as well.

### 2.5. Effective Evaluation of the Proposed Time-Sequence Model

To evaluate the effectiveness of the proposed model, univariate linear fitting was carried out in the reflected light intensity figures in a time-sequence of the nine-chip assay as well as the data in Figure 2. As demonstrated in Figure 3, the linear curve of avermectin was Y = −2.3617X − 28.3082 (Figure 3A), and a correlation coefficient of 0.9040 was obtained. It was apparent that the tested samples were distributed around the plotline with a 95% limit of agreement. Among the tested samples (Figure 3B), all the residuals bars involved the “0” point, and therefore, the produced curve could satisfyingly represent the original figures of avermectin. The linearity curve for phorate was Y = −3.9689X − 483.0526 (Figure 3C), with a correlation coefficient of 0.8093. Among the tested samples, one of the ten greatly deviated from the linearity curve, which could be purged as a false sample due to the inaccurate operation or undesirable assay conditions, while the remaining nine samples were all satisfactory (see Figure 3D). The linearity curve representing imidacloprid is shown in Figure 3E: Y = −1.6525X − 139.7500, with a satisfying correlation coefficient of 0.9635. Although the residuals bar of the tested No. 3 sample in Figure 3F did not involve the “0” point, which made it an abnormal value, the rest of the samples showed great correlation with the plot curve.

### 2.6. Sensitivity Evaluation of the Proposed Time-Sequence Model

Among the abundant chemicals, some were highly similar. Ivermectin and doramectin are both from the avermectin class, and thus, their health impacts and effects are often analyzed in pairs [31,32]. Therefore, 0.01 mg/Kg of avermectin, doramectin, and ivermectin solution was produced. Then, the prepared solution was introduced to the seven-layer chip, respectively. Each kind of chemical was introduced to the nine chips, then, the reflected light intensity of the 15 min time sequence was recorded. The recorded figures of the nine chips in each min were averaged, leading to the average optical graphic representing the general pattern of the chemical. Next, the reflected light of whole wavelengths in each min was used to identify the chemicals. The identification results of the proposed chip are shown in Figure 4. It illustrates the sensitivity of the proposed platform in distinguishing highly similar chemicals, because the tested figures were gathered by genera, and the long distance between different chemicals made them easily classifiable. It should be noted that two samples marked “X” in the ivermectin group dramatically deviated from the other eight ivermectin samples without showing a relationship with each other. If the proposed chip can be fabricated in a streamline, the structure and quality of the chip can be improved, which can make the tested figures more satisfying and the interruption of manual operation can be further reduced. Thus, it is feasible to make the “X” sample noticeably reduced. Besides, these “X” samples can be easily identified as false figures when using mathematical function modelling.

### 2.7. Real-World Analysis and Comparison with Conventional Methods

At this part, to evaluate the samples extracted from different matrices, vegetable (lettuce) and crop (rice) samples were sprayed with phorate (0.01 mg/Kg), avermectin (0.01 mg/Kg), and imidacloprid (0.5 mg/Kg), respectively. Then, each pesticide solution extracted from the crop samples and rice samples were tested using procedures similar to the nine-chip assay.

In Figure 5A, the lettuce samples sprayed with avermectin and phorate showed a high extent of convergence, while in the imidacloprid test points, two of them marginally deviated from the other seven imidacloprid tests points, and the remaining one randomly distributed while retaining the ability not to be mis-classified as with the other two pesticides due to the distance among them. The results of the rice samples were more desirable, in Figure 5B, almost all the tested samples were tightly gathered according to the pesticide used, except for only one sample from the imidacloprid group. Among all the tested figures, the abnormal samples all occurred in one specific kind of pesticide group, and manual operations and the solution preparation process might be the main reason.

Therefore, there is no doubt that the proposed method can attach a corresponding optical identification label to the specific pesticide via reflected light intensity in a time-sequence, which achieves efficiency and acceptable accuracy in pesticide identification. Compared with traditional methods, like GC-MS [33] and HPLC [22], these methods provide higher accuracy (with *R*^2^: 0.9900–1.0000) and selectivity in pesticide detection. However, food safety control is a public issue which cannot be restricted by a laboratory environment and trained technicians. These GS-MS, HPLC, and other similar methods require sophisticated apparatuses and complex chemical pretreatment [34,35]. The proposed method retains the advantages of being straightforward in operation, inexpensive, and readily accessible in materials just like the rapid pesticide test strips based on enzyme inhibitors. More importantly, it achieves the optical spectrum on-chip recording, and subsequently, sequences the recorded optical spectrum to distinguish the pesticide genre, which can alert to highly toxic chemicals before deteriorating human health. Additionally, the enzyme inhibition reaction can react automatically with ideal temperature in this microfluidic platform.

## 3. Materials and Methods

### 3.1. Design of Microfluidic Chip

Figure 6 demonstrates the schematic layout of the proposed pesticides identification platform. This paper-based microfluidic chip was composed of seven layers, which were successively arranged from top to bottom and connected end-to-end to form the central sample-flowing channel. Between the layers, water-proof double-sided adhesive tape was used to connect the layers. The inlet was fabricated in the inner circle of the first layer. The overall layout constructed a central channel inducing the sample from the inlet. The AChE (enzyme) was fixed on the second layer of the paper-based chip, which was followed by three identical layers (3rd, 4th, 5th), all of which were hollowed in the inner circle to enable the full mixture of enzyme and pesticides on the second layer, leading to the ideal inhibition of the enzyme. The indophenol acetate (substrate of the reaction) was fixed on the 6th layer, and subsequently, the color zone (7th layer) could show the chromogenic signals. As shown in Figure 6, the outer rings (light blue) were fabricated in each layer, which was fabricated by wax-printing to establish the hydrophobic barrier.

### 3.2. Enzyme-Inhibitor Based Determination

In this paper, the AChE acted as the enzyme to catalyze the hydrolysis of indophenol acetate. As demonstrated in Figure 6, firstly, the sample was introduced by the inlet fabricated in the first layer, and subsequently, due to the capillary force of the paper fiber, the solution infiltrated into the second layer of the chip. In this layer, the AChE could be fully mingled with the aquatic solution because of the three successively hollowed layers, which were designed to retain the solution in the second layer for an ideal length of time. Then, the sample came into the sixth layer and fixed with the indophenol acetate. With the presence of pesticide (inhibitor), the activity of AChE could be greatly inhibited, resulting in the incomplete hydrolysis of indophenol acetate, which made the color layer (7th) show up as a white or light-blue circle, while the absence of inhibitor enabled the full hydrolysis of the chemical reaction, as shown in Figure 6 and Formula (1), and thus, generated the acetic acid as well as indophenol (blue). The chromatic signals were detected and analyzed in the 7th layer to avoid the chromatic interference of the indophenol acetate, which was brown yellow. Therefore, the reflected light intensity spectrum in the time sequence could be obtained from the color zone precisely and automatically, and the variations of figures enabled us to establish the digital optical identification system for different pesticides.
(1)$$C_{14}H_{11}{\mathrm{NO}}_{3}+H_{2}O\overset{\;\mathrm{AChE}\;}{\to }{\;C}_{12}H_{9}{\mathrm{NO}}_{2}+{\mathrm{CH}}_{3}\mathrm{COOH}$$

### 3.3. Principle of Identification by Chromogenic Molecules

The enzyme inhibitor assay by AChE can produce a blue-colored dye (C_12_H_9_NO_2)_. It is worth mentioning that the absorbance and the time in enzyme inhibition assay are in a linear relationship [30]. In universal knowledge, the chemical reaction can be influenced by the concentration of aquatic solution and substrate [36], which involves different van der Waals interaction forces and molecular structure in the chemical reaction, and in turn, shows the variously inhibited hydrolysis efficiencies of the substrate (indophenol). Thus, the different numbers of molecules and activities at the microscale can be captured in a specific time. Figure 7 depicts various pesticides (avermectin, imidacloprid, and phorate) with their molecular structures. Although all of them can be deemed as inhibitors of AChE, at the molecular scale, these pesticides are different in interactions with the enzyme and van der Waals interaction within the molecule due to the diverse molecular pattern [36]. Therefore, when these disparate pesticides react with the same AChE, distinct inhibition rates occur, and during the same period of time, different numbers of AChEs and molecules are inactivated by each genre of pesticide. Therefore, via the captured sequence time model of reflected light intensity by the proposed platform, it is feasible to establish an identification system.

### 3.4. Pesticide Distinguish Assay

In this paper, the optical fiber used was QP600-2-VIS-NIR (OceanView, America) and the light came from the optical source HL-2000-FUSE (OceanView, America). If the optical fiber and source were correctly connected, then the captured reflected light intensity was recorded on the computer using Oceanview software. Phorate (3%), imidacloprid (10%), and avermectin (0.9%) were purchased from Kunshan Jialong Biotechnology Co., Ltd. Distilled water and other chemicals were of analytical standard. Then, the pesticides were resolved in the obtained distilled water to produce the aquatic solution of diverse concentration. Figure 8 illustrates the whole process of a single proposed identification method of pesticides. Firstly, a total of nine microfluidic paper-based chips were placed on the constant thermal heating device (GDANA HT-300) (37 °C) to achieve the ideal reaction temperature of AChE. After 2 min, the sample (30 μL) was introduced to the paper-based chip via an inlet fabricated in the first layer with an interval of the 30 s. When the first chip was introduced to the sample, the time counting began. Therefore, the reflected light intensity of the 1st, 2nd, 3rd, 4th, 5th, 6th, 7th, 8th, and the 9th chip was tested after the 60 s, 90 s, 120 s, 150 s, 180 s, 210 s, 270 s, 330 s, and 390 s, respectively. Then, the optical spectrum of the first min was recorded by optical devices (Ocean Optics USB2000+, America), the reflected light intensity was automatically recorded by the optical fiber placed underneath the chip. For the light intensity of *i* min in Figure 8, at (*i* × 60 + (*No*. – 1) × 30) s after the sample was introduced at the inlet of the No. 1 chip, the figures for reflected light intensity were recorded. This was looped until *i* exceeded 15, and thus, the optical spectrum of 15 min was recorded. During this period, different enzyme activities occurred according to pesticide concentration and molecules. Error figures which showed a dramatic difference with other chips were deemed as noise, and thus, deleted from the results. For this purpose, at the beginning of the digital processing, multiplicative scatter correction was used to purge the noise from the optical spectrum figures recorded from 118 nm to 1112 nm, and subsequently, through principal component analysis the 2047 dimension figures became 20 dimension figures while retaining the major information. Then, the fluctuations in the entire optical spectrum could be reduced, as the significant change that had taken place in the reflected light intensity spectrum of the chip was mainly aroused by the enzyme inhibition activity and the corresponding chromogenic compounds. Thus, the impact of the matrix effect on the identification was limited after the algorithm was used, because for one thing, the entire reflected light intensity spectrum (118 nm to 1112 nm) was utilized in this assay. For another thing, the principal component analysis and multiplicative scatter correction greatly eliminated the limited reflected light intensity caused by other chemicals, and other chemicals (except most Ops and Cps) will not inhibit the hydrolysis of indophenol, which makes the chromogenic modules mainly come from the substrate (indophenol) fixed in the chip. This phenomenon can also be observed in Figure 1; the reflected light intensity changed noticeably min by min in the 15 min assay, which was much more noticeable than the light intensity changes of other chemicals might occur in the aquatic solution. Then the collected figures of the nine chips for each min were averaged during the 15 min period. After the filtration of noise and dimension reduction, the mathematical model of the reflected light spectrum by time-sequence could be established. Moreover, according to specific time-arrayed reflected light spectrum, intelligent devices can attach to the pesticide with a specific optical ID card. This achieves the prewarning of food health control, preventing the highly ztoxic chemicals from being taken into the human body.

## 4. Conclusions

A first of its type, a seven-layer paper-based pesticide identification microfluidic chip was proposed in this paper, which enabled the digitalization of optical information and was equipped with the ability to distinguish each pesticide by a unique time-sequence optical spectrum. Three kinds of pesticides, namely, phorate, avermectin, and imidacloprid, as well as avermectins were analyzed by the proposed chip, and then, the specific concentration of three pesticides were sprayed on real-world samples to test the effects. The results indicated good selectivity and sensitivity, and thus, the proposed platform can successfully identify those pesticide. In this assay, the materials were readily accessible and the equipment was simple to use. Therefore, the proposed platform paves a potential way for ubiquitous health control under non-laboratory-based settings, while providing a new way to label identification information for different kinds of pesticides via enzyme inhibition assays.

## Figures and Tables

**Figure 1 molecules-24-02428-f001:**
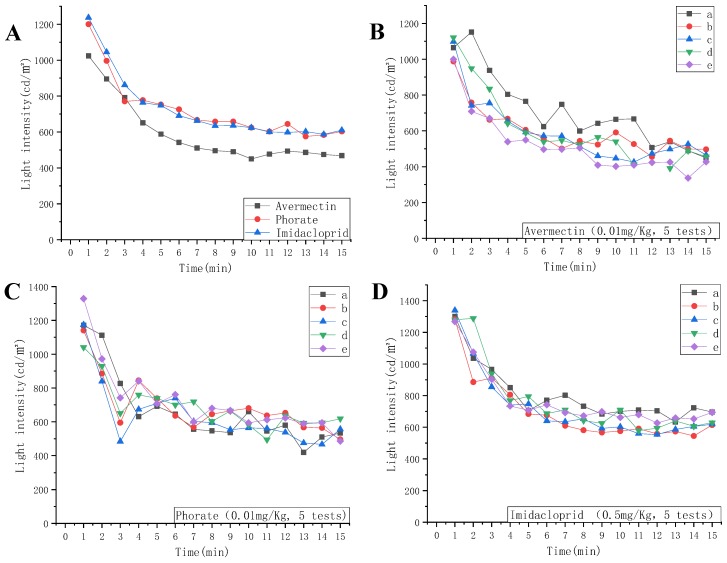
The light intensity change model of the three pesticides. (**A**) The average light intensity (wavelength: 611.59 nm) of each min by the 9 proposed test chips during a period of 15 min, with a concentratiom of 0.5 mg/Kg in the imidaloprid sample and 0.01 mg/Kg for the phorate and avermectin samples. (**B**) The light intensity of each min for 5 competitive tests using the proposed chip for 0.01 mg/Kg of avermectin, (**C**) 0.01 mg/Kg of avermectin, and (**D**) 0.5 mg/Kg of imidaloprid. Each color line records the reflected light intensity of each minute by one chip, and all of the five chips (lines) in each figure are tested by the same concentration demonstrated at the bottom of the figure.

**Figure 2 molecules-24-02428-f002:**
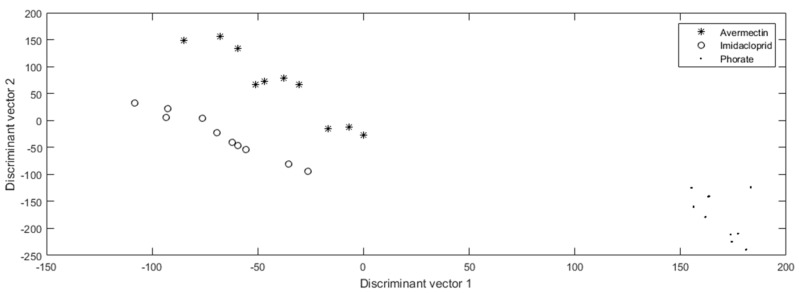
The pesticide identification results of phorate (0.01 mg/Kg), avermectin (0.01 mg/Kg), and imidacloprid (0.5 mg/Kg).

**Figure 3 molecules-24-02428-f003:**
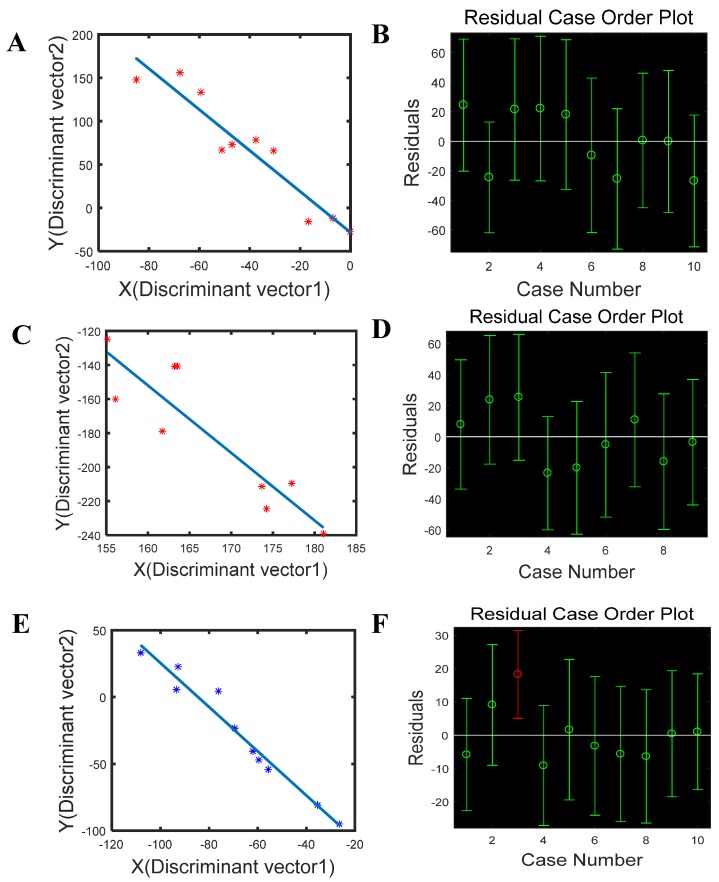
The univariate linear fitting results of (**A**) avermectin and (**B**) its tested residuals results, (**C**) phorate and (**D**) its tested residuals results, as well as (**E**) imidacloprid and (**F**) its tested residuals results.

**Figure 4 molecules-24-02428-f004:**
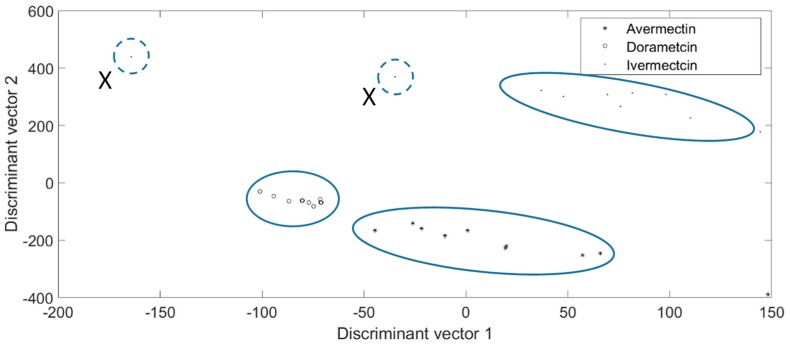
The sensitivity evaluation results of the proposed chip tested by in 0.01 mg/Kg of avermectins. The figures are gathered by groups and the distance between different chemicals is distant.

**Figure 5 molecules-24-02428-f005:**
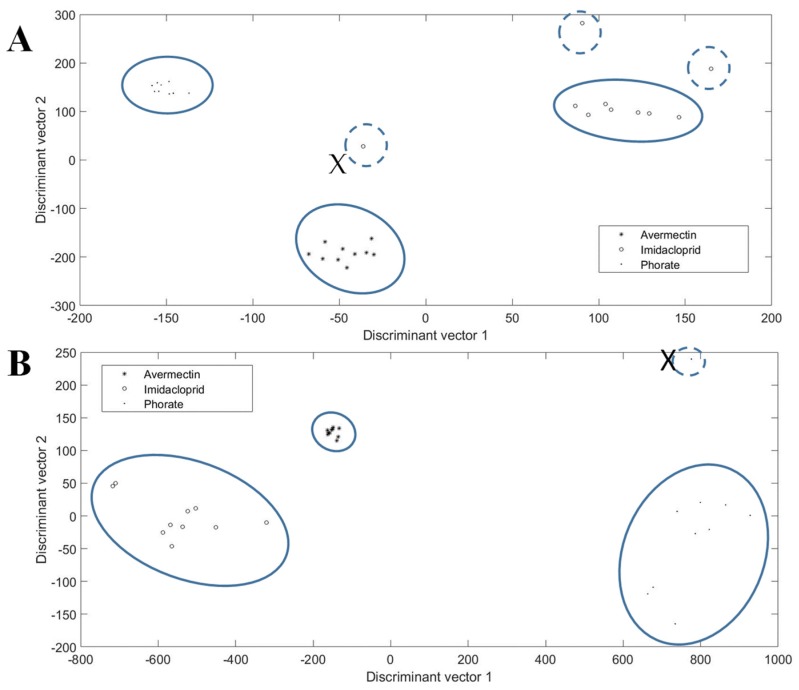
The cluster analysis results of phorate (0.01 mg/Kg), avermectin (0.01 mg/Kg), and imidacloprid (0.5 mg/Kg) sprayed on real-world samples of (**A**) lettuce and (**B**) rice.

**Figure 6 molecules-24-02428-f006:**
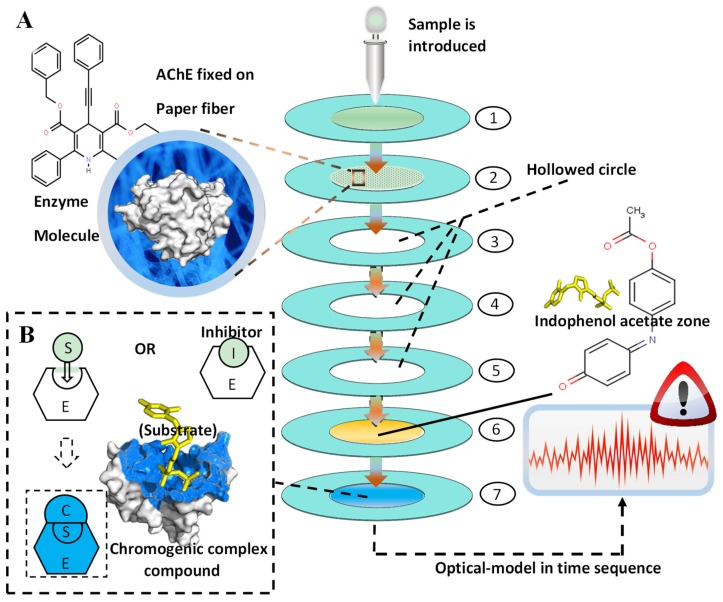
The schematic overview of the proposed pesticide identification platform. (**A**) The layout of the seven-layer microfluidic paper chip and the molecular structure of AChE and indophenol acetate (substrate). After the chromatic hydrolysis of indophenol acetate via the chip, the absorbance figures were processed and analyzed by intellectual devices. (**B**) The chemical reaction of enzyme inhibition on the molecular scale. With the absence of inhibitor (Organophosphate pesticides and carbamate pesticides), the S (substrate) could be catalyzed by AChE, fabricating the colored complex compound, which could be introduced to the 7th layer of the proposed chip.

**Figure 7 molecules-24-02428-f007:**
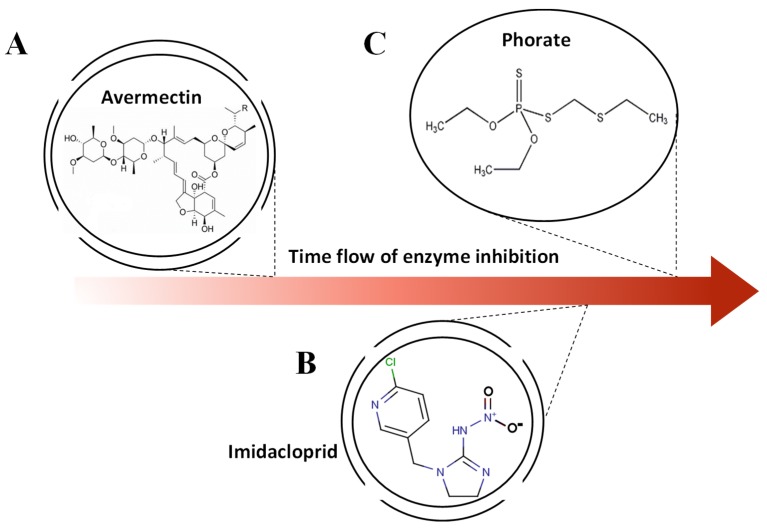
The various molecular structures of (**A**) avermectin, (**B**) imidacloprid, and (**C**) phorate and the aroused different efficiency in enzyme inhibition.

**Figure 8 molecules-24-02428-f008:**
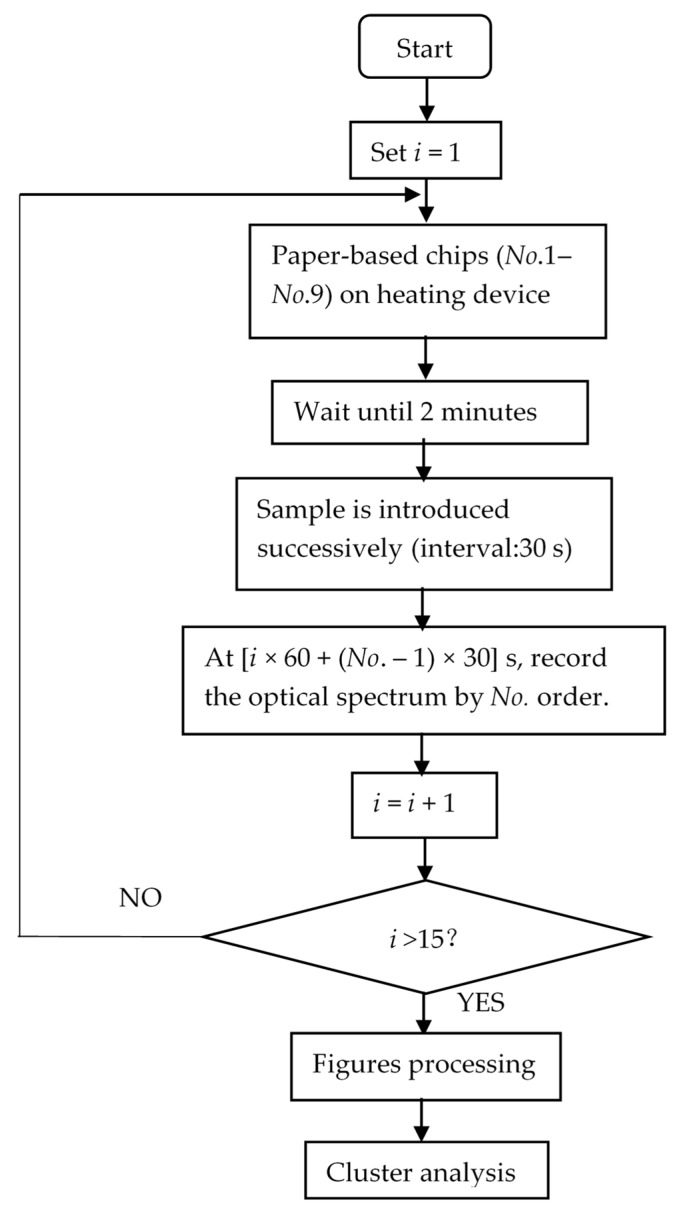
The flow chart of the pesticide identification assay via absorbance model and chromogenic molecules.

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
