# Peer review of "Pesticide Residues Identification by Optical Spectrum in the Time-Sequence of Enzyme Inhibitors Performed on Microfluidic Paper-Based Analytical Devices (µPADs)"

_molecules, 2019, doi:10.3390/molecules24132428_

Round 1

Reviewer 1 Report

General comments:

In this manuscript the authors sought to develop a 7-layer paper-based pesticide identification microfluidic chip for the identification of phorate, avermectin and imidacloprid in the aquatic solution. In view of this study, the authors showed that the linear discriminant analysis shows satisfying discrimination of the imidacloprid, phorate, and avermectin. The correlation coefficient for theses linearity curve is 0.9635, 0.8093, and 0.9094 respectively with a 95% limit of agreement. Overall it is a passably written manuscript that it is feasible to distinguish the tested three kinds of pesticides by the change of reflected light spectrum in each minute via the proposed chip. However, there are some points the need further clarification as follows.

Abstract: this part is insufficient, especially in matrix effect. It is need to be supplementary.

Introduction: the description should be written more concisely. The spaces of Line 43-86 are occupied too much spaces and too verbose. Additionally, I suggest the authors need to focus the chip analyze in the identification of pesticides and supplement the related information.

Materials and methods:

The authors need to express whether the different chemicals may affect the identification of pesticides. Furthermore, I suggest the authors should explain the matrix effect.

The problems may be associated with relatively high similar chemicals of avermectin classes, such as abamectin, doramectin, eprinomectin, emamectin, ivermectin and moxidectin. And at this point, we can see the importance of the sensitivity of the method used. Please explain and provide more information that the chip can distinguish the avermectin from avermectin classes.

Results and discussion:

The matrixes from different vegetable, or crop may affect the identification of phorate, avermectin and imidacloprid or not. The authors need to clearly describe and add some information to express the matrix effect.

The authors need to explain these tested levels could be suitable for all different matrixes (phorate at 0.01 mg/Kg, avermectin at 0.01 mg/Kg and imidacloprid at 0.5 mg/Kg). Moreover, the authors need to discuss the effect of those chemicals in the results of method validation.

The discussion need more information to show the significant among different studies or references. In addition, the authors need to add more information to compare the different analytical methods, such as MS/MS.

Conclusions: these descriptions of this part is not different with results or discussion. I am not sure the authors wanted to show the “Results”, “Discussion” or “Conclusions”.

Author Response

Point 1: Abstract: this part is insufficient, especially in matrix effect. It is need to be supplementary

Response 1: See the revised description: After pretreatment of figures of reflected light intensity during the 15-minute period, the figures mainly focus on the reflected light variations aroused by enzyme inhibition assay….’.  Other chemicals might impact the reflected light spectrum observed, but the complex chromogenic compounds by enzyme inhibition assay or the inhibition activity are the main reason for the change of optical graphic. During the 15-min experiment, from the Figure 1, we can see the apparent change of reflected light, and therefore via the pretreatment of optical figures and principle component analysis, other trivial optical change by other chemicals are limited.

Point 2: Introduction: the description should be written more concisely. The spaces of Line 43-86 are occupied too much spaces and too verbose. Additionally, I suggest the authors need to focus the chip analyze in the identification of pesticides and supplement the related information.

Response 2: See the revised description at ‘Introduction’. We delete some verbose description, and clearly named some researches that have some advantages in the pesticide analysis and quantification. The van der Waals forces and interactions between molecules and the molecule property is important in this paper. So, the fluorescent analysis of pesticide by Quantum dots (QDs)-based molecularly imprinted polymer (MIP) composite nanospheres proposed by Zhao yaoyao et al. is implemented to the reference. At the third paragraph of introduction, we implement the description as follows: ‘Optical spectroscopy can indicate the radiation loss of energy……various optical figures if optical spectroscopy is used to track the whole chemical reaction.’. We add some information to indicate the relationship of optical spectroscopy with the forces and activity between and within molecules. To show the principle why the chip can analyse and identify the different pesticides.

Point 3: Materials and methods:The authors need to express whether the different chemicals may affect the identification of pesticides. Furthermore, I suggest the authors should explain the matrix effect

Response 3: At 3.4 pesticide distinguish assay, we revise the description as follows:’ For this purpose, at the beginning of the digital processing, multiplicative scatter correction is used to purge the noise….. which makes the chromogenic modules mainly come from the substrate (indophenol) fixed in the chip.’ The chemical other than Ops and Cps will not react with the enzyme and thus inhibit it. Therefore, the change of reflected light intensity mainly comes from the activity of enzyme inhibitors or the produced chromogenic compounds. Moreover, this phenomenon is also apparent in the revised Figure 1, with time flow of the 15-minute inhibition assay, the reflected light intensity changes minute by minute, which is much more intensity than the impacts that might occur by other chemicals in this assay. In addition, the figures processing utilises some algorithm methods to purge the noises and extracts the principle information (principle component analysis). At last, we also used the whole optical spectrum with 2047-dimension figures to alleviate the situation that some chemicals might have specific absorption peak at some wavelengths.

Point 4: The problems may be associated with relatively high similar chemicals of avermectin classes, such as abamectin, doramectin, eprinomectin, emamectin, ivermectin and moxidectin. And at this point, we can see the importance of the sensitivity of the method used. Please explain and provide more information that the chip can distinguish the avermectin from avermectin classes.

Response 4: In order to better clarify this problem, we implemented 2.6 section: sensitivity evaluation of the proposed time-sequence model. See descriptions as follows:’ Among the abundant ….can be easily identified as false figure when mathematical function modelling. At this part, apart from two samples marked with ‘X’ in ivermectin groups, other chemicals show highly extent of convergence while distributing far from other type of chemical group. Thus, the proposed method is able to distinguish those highly similar chemicals. The results are imaginable, as this assay mainly based on inhibition efficiency and different molecule interactions due to different forces and structure within molecules, this reaction principle is already equipped with highly extent of sensitivity. Apart from this, among all the samples tested in this paper, the ‘X’ sample all distribute in one group, which shows that this can be largely due to the problems in tge fabrication of the specific tested chip or the sample preparation process.  If the chips are fabricated in streamline, such problems are feasible to greatly alleviated.

Point 5: Results and discussion: The matrixes from different vegetable, or crop may affect the identification of phorate, avermectin and imidacloprid or not. The authors need to clearly describe and add some information to express the matrix effect

Response 5: In order to better clarify this problem, we implemented 2.7 section: real-world analysis and comparison with conventional methods. We choose one type of vegetable (lettuce) and one type or crop (rice), and sprayed them by the tested three chemicals with the same concentration. See descriptions :’ vegetable (lettuce) and crop (rice) samples are sprayed with phorate (0.01mg/Kg), avermectin(0.01mg/Kg), and imidacloprid (0.5 mg/Kg) respectively. Then each pesticide solution extracted from the crop samples and rice samples are tested by similar procedures of the nine-chip…..almost all the tested samples tightly gathered according to pesticide used except just one sample of the imidacloprid group. ‘ The performance of the assay performed on real-word samples are also good, although some sample deviates from its sample groups, they are demanding to be falsely recognized as the other two chemicals.

Point 6: The authors need to explain these tested levels could be suitable for all different matrixes (phorate at 0.01 mg/Kg, avermectin at 0.01 mg/Kg and imidacloprid at 0.5 mg/Kg). Moreover, the authors need to discuss the effect of those chemicals in the results of method validation

Response 6: See the description:’ In China, the maximum residue level (MRL) for imidacloprid is 0.5 mg/Kg, while the MRL value for highly toxic pesticides like phorate or avermectin is both 0.01 mg/Kg.’ The reason why this assay utilises the assigned MRL of Chinese government is that the MRL is mainly based on the different inhibition rate of those chemicals, and the MRL for European is quite similar. More importantly, the proposed method is not solely for optical identification, it also gets the visual detection and analysis function just like other rapid pesticide detection strip, as they are all based on enzyme inhibition assay. The difference lies in optical spectrum on-chip identification of the method described in this paper, and thus we did not put the attention on its detection of limit and things like that, otherwise the structure will be too verbose.

Point 7: The discussion need more information to show the significant among different studies or references. In addition, the authors need to add more information to compare the different analytical methods, such as MS/MS.

Response 7:  In order to clarify this point, we revised and implemented 2.7 section: Real-world analysis and comparison with conventional methods. See description as follows: ‘Therefore, there is no doubt that the proposed method can attach corresponding optical identification label to the ….. those highly toxic chemicals can be prewarned before deteriorating human health. Additionally, the enzyme inhibition reaction can react automatedly with ideal temperature in this microfluidic platform.’  In conclusion, the GC-MS or HPLC and other conventional methods no doubt provide higher sensitivity and accuracy. While the requirements of sophisticated apparatus and laboratory-based operation conditions, long time sample pretreatment and trained technicians makes these problems undesirable in low-resource settings. In addition, in this region, farmer might pay much more attention to the effects of the pesticide in order to promote the crop yield while ignoring the potential health impacts. This method uses the readily accessible material (paper) to build the whole pesticide identification method. The cost is low, and the 7-layer structure makes the sample introduction and reaction highly automated. Moreover, the chip gives each pesticide with its own optical identification, and thus the consumers and governments can easily found those food polluted by pesticides with high poisonousness.

Point 8: Conclusions: these descriptions of this part is not different with results or discussion. I am not sure the authors wanted to show the “Results”, “Discussion” or “Conclusions”

Response 8:  The former conclusion part includes too much unnecessary information, and therefore we add more conclusive description and delete some words. See the revised description:’ A first of its type 7-layer paper-based pesticide identification microfluidic chip is proposed in this paper, …. under nonlaboratory-based settings, while gives a new way to label the identification information to different kinds of pesticides by enzyme inhibition assays.’

Reviewer 2 Report

The paper is good for the publication in the journal.

The author should implement the references with the actual paper on the residue analysis as anastassiades et al in particular in the introduction when the authors describes the analytical technique for the analysis of pesticides (line 43-49).

The figure 4 B, 4C, 4D should report the legend of the lines.

Author Response

Point 1: The author should implement the references with the actual paper on the residue analysis as anastassiades et al in particular in the introduction when the authors describes the analytical technique for the analysis of pesticides (line 43-49).

Response 1: We revised the introduction part with regard to this problem. See descriptions as follows: ‘Ye J et al. (2009)… achieve the multi-analysis of several pesticides.’ ‘With regard to other methods, Vetrova E et al. (2007) used bioluminescent signal system to indicate the toxicants like organophosphates [11]. Zhao Y et al. (2012) successfully used the interaction including van der Waals forces … leading to the undesirable precision level’. The description is revised to be more specific and more detailed.

Point 2: The figure 4 B, 4C, 4D should report the legend of the lines.

Response 2: As for these three diagrams, they are all tested by the same concentration of chemical, to avoid the confusion which might occur to the readers. We revised the former figure 4 ( Figure 1 at the revised edition). The tested concentration and the genre of the chemicals are implemented to the legend.

Reviewer 3 Report

The section of materials and methods could be more specific in how the detection method works, and mention the specification of equipment and software employed.

Figure 1 is difficult to appreciate, put in order the objects in the figure and maybe separate the microfluidic chip’s structure and the reactions A and B in two different figures.

Line 129: It is worth mentioning that the absorbance and the time in enzyme inhibition assay is are in linear relationship.

Line 130 Try to explain in a different way the dependence of the reaction rate. The idea is confuse.

Line 135: if you are talking about energy, discuss that data and present the information.

Line 150: what is the sample volume?

Section 3.1 The selection of wavelength: I recommend you to include a graphic with the absorbance of the solution, this gives you the exact information of the best working wavelength, instead of just saying the wavelength is around 600 nm. Also you are talking about sensitivity, what is the sensitivity of your detection method? Maybe include a table with this information in order to compare the results in an easier way.

Section 3.3 Repetitive assays: I suggest you to explain better the reason for the behavior of the three pesticides on the light intensity graphics, the smoothness and the fluctuations respectively. Why does this happen?

The manuscript is interesting with quality content, however English grammar revision is important, because of the ideas can be unclear.

It is important to follow the Journal template. Introduction, Results, Experimental, Conclusions and References.

Author Response

Point 1: The section of materials and methods could be more specific in how the detection method works, and mention the specification of equipment and software employed.

Response 1: At 3.4 pesticide distinguish assay, we add the basic information about the fiber, light source and the software used. See descriptions:’ In this paper, the optical fiber used is QP600-2-VIS-NIR (OceanView, America),….by using Oceanview software.’ As for how the detection works, we add more details about which algorithm is used to purge the noise, and how the principle information of the spectrum is correctly extracted. Meanwhile, in the introduction part, we add more information about the principle of this method, and in the revised experimental part, more description about the method is implemented.

Point 2: Figure 1 is difficult to appreciate, put in order the objects in the figure and maybe separate the microfluidic chip’s structure and the reactions A and B in two different figures.

Response 2: The previous Figure 1 is a little bit chaos, therefore in the revised Figure 6 (the previous Figure 1), we clearly discriminate how the enzyme (AChE) is inhibited by the pesticides, and how hydrolysis product of the indophenol (substrate) with an area marked with Figure 6B. Hope this revision will make this figure more understandable.

Point 3: Line 130 Try to explain in a different way the dependence of the reaction rate. The idea is confuse.

Response 3: See descriptions at 3.3:’ the chemical reaction can be influenced by the concentration of aquatic solution and … different number of molecules and activity in microscale can be captured in specific time.’ We change the reaction rate with some specific words in order to be better understandable to readers, and the difference of the reaction mainly comes from the different molecule interactions and the van der Waals interaction forces. Therefore, we focus the description on this.

Point 4: Line 135: if you are talking about energy, discuss that data and present the information.

Response 4: The different energy mainly involves the different molecule structures, and thus the van der Waals interaction forces and the hydrolysis efficiency of the substrate is variously inhibited. In order to better clarify this point, the descriptions are revised at the 3.3 part.

Point 5: Line 150: what is the sample volume?

Response 5: The tested sample volume is 30μL, which is implemented to the part 3.4.

Point 6: Section 3.1 The selection of wavelength: I recommend you to include a graphic with the absorbance of the solution, this gives you the exact information of the best working wavelength, instead of just saying the wavelength is around 600 nm. Also you are talking about sensitivity, what is the sensitivity of your detection method? Maybe include a table with this information in order to compare the results in an easier way

Response 6: In order to clarify this, we implement the supplementary figure 1, and it is obvious that the reflected light intensity as 611.59nm is the greatest. Thus, at this wavelength, the change aroused by the enzyme inhibition is also the greatest. It is worth mentioning that in the cluster analysis and principle component analysis, the whole optical spectrum is used, other than the figure just at this point. The sensitivity assay is implemented at part 2.6. The proposed chips are analysed with highly similar chemicals, and the produced results show that the platform can identify the chemicals by genre. See descriptions as follows:’ Ivermectin and doramectin are both from the avermectin class, and thus their health impacts and effects often analyzed in pairs …..different chemicals makes them can be easily classified.’

Point 7: Section 3.3 Repetitive assays: I suggest you to explain better the reason for the behavior of the three pesticides on the light intensity graphics, the smoothness and the fluctuations respectively. Why does this happen?

Response 7: With the process of enzyme inhibition assay, the hydrolysis of the substrate produce some chromogenic compounds, compared with the previous white circle, the chromogenic area can absorb more light, and thus reduce the reflected light. With the time flow, the enzyme inhibition activity and the hydrolysis of indophenol remain constant, and thus the smooth graphic pattern appears. While the produced chromogenic modules of phorate can be unstable at the previous minutes, while makes the reflected light intensity just reduce and then increase.

Round 2

Reviewer 1 Report

These authors had revised the new vision.